# PixMIM:
# Rethinking Pixel Reconstruction in Masked Image Modeling

**Yuan Liu**\*                                          *seuyou2333@gmail.com*
*Shanghai AI Laboratory*

**Songyang Zhang**\*                                  *zhangsongyang@pjlab.org.cn*
*Shanghai AI Laboratory*

**Jiacheng Chen**                                      *cjc0722hz@gmail.com*
*Simon Fraser University*

**Kai Chen**†                                          *chenkai@pjlab.org.cn*
*Shanghai AI Laboratory*

**Dahua Lin**                                          *lindahua@pjlab.org.cn*
*Shanghai AI Laboratory*
*The Chinese University of Hong Kong*

**Reviewed on OpenReview:** *https://openreview.net/forum?id=qyfz0QrkqP*

## Abstract

Masked Image Modeling (MIM) has achieved promising progress with the advent of Masked Autoencoders (MAE) and BEiT. However, subsequent works have complicated the framework with new auxiliary tasks or extra pre-trained models, inevitably increasing computational overhead. This paper undertakes a fundamental analysis of MIM from the perspective of pixel reconstruction, which examines the input image patches and reconstruction target, and highlights two critical but previously overlooked bottlenecks. Based on this analysis, we propose a remarkably simple and effective method, PixMIM, that entails two strategies: 1) filtering the high-frequency components from the reconstruction target to de-emphasize the network's focus on texture-rich details and 2) adopting a conservative data transform strategy to alleviate the problem of missing foreground in MIM training. PixMIM can be easily integrated into most existing pixel-based MIM approaches (*i.e.*, using raw images as reconstruction target) with negligible additional computation. Without bells and whistles, our method consistently improves four MIM approaches, MAE, MFF, ConvMAE, and LSMAE, across various downstream tasks. We believe this effective plug-and-play method will serve as a strong baseline for self-supervised learning and provide insights for future improvements of the MIM framework. Code and models will be available.

## 1 Introduction

Recent years have witnessed substantial progress in self-supervised learning (SSL). Inspired by the success of masked language modeling (MLM) in language processing, masked image modeling (MIM) has been introduced to computer vision, leading to rapid growth in SSL. Pioneering works such as BEiT (Bao et al., 2021) and MAE (He et al., 2022) exploit Vision Transformers (ViT) to learn discriminative visual representations from raw image data without manual annotations, and their transfer learning performance has outperformed the supervised learning counterpart.

---

\* Contributed equally. † Corresponding author.

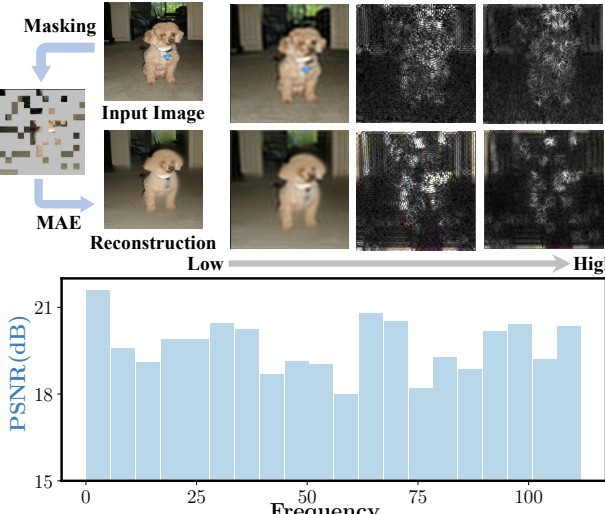

Figure 1: **Frequency analysis with an example image.** (Top) Components belonging to different frequency intervals for the input and MAE-reconstructed image. (Bottom) The peak signal-to-noise ratio (PSNR) of the reconstruction for components of fine-grained frequency intervals. MAE tends to focus on both the low and high-frequency components and reconstructs intricate details. We increased the image brightness for better visualization.

Early MIM methods share a simple pipeline – a portion of non-overlapped image patches are randomly masked, and the model learns to extract discriminative representations by reconstructing the pixel or feature values of the masked patches (Bao et al., 2021; He et al., 2022; Xie et al., 2022). To improve the representation quality, some advanced MIM works (Huang et al., 2022; Zhou et al., 2022) incorporate extra auxiliary tasks (*e.g.*, contrastive learning) while some other efforts leverage powerful pre-trained models for distillation (Hou et al., 2022; Peng et al., 2022a). However, these attempts either complicate the overall framework or inevitably introduce non-negligible training costs.

Unlike recent works, this paper investigates the most fundamental but usually overlooked components in the data reconstruction process of MIM, *i.e.*, *the input image patches and the reconstruction target*, and proposes a simple yet effective method that improves a wide range of existing MIM methods while introducing minimal computation overhead. The core of the paper is a meticulous analysis based on the milestone algorithm – MAE (He et al., 2022), which discloses critical but neglected bottlenecks of most pixel-based MIM methods. The analysis yields two important observations:

**(1). Reconstruction target**: Since the advent of MAE, most MIM methods have adopted raw pixels as the reconstruction target. The training objective requires perfect reconstruction of the masked patches, including intricate details, *e.g.*, textures. This perfect reconstruction target tends to waste the modeling capacity on short-range dependencies and high-frequency details (see Figure 1), which has been pointed out by BEiT (Bao et al., 2021) and also broadly studied in generative models (Ramesh et al., 2021; Rombach et al., 2022). In addition, studies on the shape and texture bias (Geirhos et al., 2019; 2021) indicate that models relying more on shape biases usually exhibit better transferability and robustness. However, reconstructing the fine-grained details inevitably introduces biases toward textures, thus impairing the representation quality.

**(2). Input patches**: MAE employs the commonly used Random Resized Crop (RRC) for generating augmented images. However, when coupling the RRC with an aggressive masking strategy (*i.e.*, masking out 75% image patches), the visible patches in MAE's input can only cover 17.1% of the key object on average (see Figure 2 for examples and Section 3.2 for details). Semantic-rich foregrounds are vital for learning good visual features (Touvron et al., 2022). The low foreground coverage during training likely hinders the model's ability to effectively capture the shape and semantic priors, thus limiting the quality of representation.

Guided by the analysis, we propose a method consisting of two simple yet effective modifications to the MIM framework. Firstly, we apply an ideal low-pass filter to the raw images to produce the reconstruction

targets, such that the representation learning prioritizes the low-frequency components (e.g., shapes and global patterns). Secondly, we substitute the commonly used RRC with a more conservative image transform operation, *i.e.*, the Simple Resized Crop (SRC) employed by AlexNet (Krizhevsky et al., 2012), which helps to preserve more foreground information in the inputs and encourages the model to learn more discriminative representation. As our method operates directly on raw pixels of the input patches and reconstruction targets to improve the MIM framework, we dub it **PixMIM**. Figure 4 illustrates the overall architecture of our method.

PixMIM can be effortlessly integrated into most existing pixel-based MIM frameworks. We thoroughly evaluate it with three well-established approaches, MAE (He et al., 2022), ConvMAE (Gao et al., 2022), MFF(Liu et al., 2023), and LSMAE (Hu et al., 2022). The experimental results demonstrate that PixMIM consistently enhances the performance of the baselines across various evaluation protocols, including the linear probing and fine-tuning on ImageNet-1K (Deng et al., 2009), the semantic segmentation on ADE20K (Zhou et al., 2018), and the object detection on COCO (Lin et al., 2014), without compromising training efficiency or relying on additional pre-trained models. We additionally conduct experiments to assess the model's robustness against domain shift and exploit an off-the-shelf toolbox (Geirhos et al., 2021) to analyze the shape bias of the model, which further highlights the strength of our method. In summary, our contributions are three-fold:

- We carefully examine the reconstruction target and input patches of pixel-based MIM methods, which reveals two important but previously overlooked bottlenecks.
- Guided by our analysis, we develop a simple and effective plug-and-play method, PixMIM, which filters out the high-frequency components from the reconstruction targets and employs a simpler data transformation to maintain more object information in the inputs.
- Without bells and whistles, PixMIM consistently improves three recent MIM approaches on various downstream tasks with minimal extra computation.

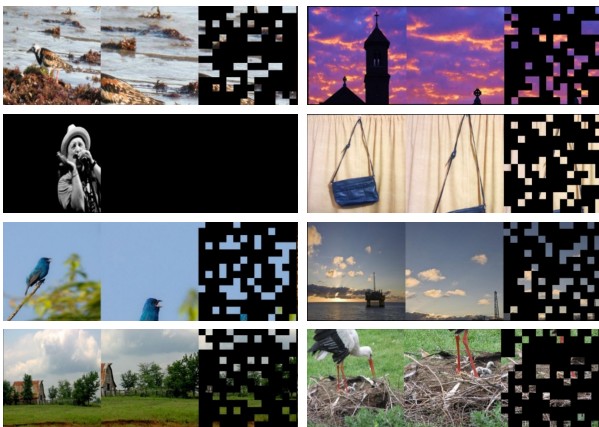

Figure 2: **Visualization of MAE's input patches.** For each example, we show the original image, the image after the Random Resized Crop (RRC), and the visible patches produced by MAE's masking strategy from left to right. The coupling of RRC and an aggressive masking strategy leads to low foreground coverage in the inputs and potentially impairs the representation quality.

## 2 Related Works

**Self-supervised Learning.** Since the success of BERT (Devlin et al., 2019) and GPT series (Radford et al., 2018; Brown et al., 2020) in natural language processing, self-supervised learning (SSL) has made revolutionary progress in various areas and gradually replaced the conventional supervised learning paradigm. One main-stream SSL framework in computer vision is contrastive learning, which has shown great effectiveness in image (Chen et al., 2020b; He et al., 2020; Grill et al., 2020; Chen & He, 2021), video (Qian et al., 2021; Feichtenhofer et al., 2021; Hu et al., 2021; Liu et al., 2022b), and multi-modal data (Radford

et al., 2021; Jia et al., 2021; Chen et al., 2021; Li et al., 2021). The main philosophy of contrastive learning is to enforce the model to pull augmented views of the same data samples together while pushing views of different samples apart, such that the model learns to extract discriminative representations.

**Masked Image Modeling.** Compared to contrastive learning, masked image modeling is another paradigm for self-supervised representation learning. It works by masking a portion of an image and enforcing the model to reconstruct these masked regions. BEiT (Bao et al., 2021) masks 60% of an image and reconstructs the features of these masked regions, output by DALL-E (Ramesh et al., 2021). Recently, there are also some attempts (Peng et al., 2022a; Hou et al., 2022) to align these masked features with features from a powerful pre-trained teacher model, *e.g.*, CLIP (Radford et al., 2021). Since these target features contain rich semantic information, the student model can achieve superior results on many downstream tasks. Instead of reconstructing these high-level features, MAE (He et al., 2022) reconstructs these masked pixel values. Besides, MAE only feeds these visible tokens into the encoder, which can speed up the pre-training by $3.1\times$, compared to BEiT.

## 3 A Closer Look at Masked Image Modeling

In this section, we first revisit the general formulation of Masked Image Modeling (MIM) and describe the fundamental components (Section 3.1). We then present a careful analysis with the milestone method, MAE (He et al., 2022), to disclose two important but overlooked bottlenecks of most pixel-based MIM approaches (Section 3.2), which guides the design of our method.

### 3.1 Preliminary: MIM Formulation

The MIM inherits the denoising autoencoder (Vincent et al., 2008) with a conceptually simple pipeline, which takes the corrupted images and aims to recover the masked content. The overall framework typically consists of 1) data augmentation & corruption operation, 2) the auto-encoder model, and 3) the target generator. Table 1 compares representative MIM methods based on the three components. Formally, let $\mathbf{I} \in \mathbb{R}^{H \times W \times 3}$ be the original image, and $H, W$ are the height and width of the image, respectively. The corrupted image $\hat{\mathbf{I}}$ is generated with augmentation $\mathcal{A}(\cdot)$ and corruption $\mathcal{M}(\cdot)$, as $\hat{\mathbf{I}} = \mathcal{M}(\mathcal{A}(\mathbf{I}))$. As in supervised learning, the random resized crop (RRC) is the *de facto* operation for $\mathcal{A}(\cdot)$ in MIM (He et al., 2022; Bao et al., 2021; Xie et al., 2022). The corruption $\mathcal{M}(\cdot)$ is instantiated by masking image patches with different ratios (*e.g.* 75% in MAE (He et al., 2022) and 60% in SimMIM (Xie et al., 2022)).

| Method | aug&corruption | auto-encoder | target |
|---|---|---|---|
| BEiT (Bao et al., 2021) | RRC+40% mask | ViT+Linear | DALLE |
| SimMIM (Xie et al., 2022) | RRC+60% mask | ViT+Linear | RGB |
| MaskFeat (Wei et al., 2022) | RRC+40% mask | ViT+Linear | HOG |
| ConvMAE (Gao et al., 2022) | RRC+75% mask | ConvViT+MSA | RGB |
| MAE (He et al., 2022) | RRC+75% mask | ViT+MSA | RGB |

Table 1: **Empirical decomposition of MIM approaches.** aug: data augmentation. mask: mask ratio. MSA: multi-head self-attention layer. RRC: random resized crop.

The reconstruction target $\mathbf{Y}$ is also a key component of MIM methods. We denote the target generator function by $\mathcal{T}(\cdot)$, and the target is produced by $\mathbf{Y} = \mathcal{T}(\mathcal{A}(\mathbf{I}))$. The community has explored various target generation strategies $\mathcal{T}(\cdot)$, which could be roughly divided into non-parametric strategies (*e.g.*, identity function for RGB, HOG) and parametric strategies (*e.g.*, a pre-trained model like DALLE (Ramesh et al., 2021) or CLIP (Radford et al., 2021)). Our analysis focuses on the non-parametric family, as it does not rely on external pre-training data and is typically more computationally efficient.

Given the target $\mathbf{Y}$, the autoencoder model $\mathcal{G}(\cdot)$ takes the corrupted images $\hat{\mathbf{I}}$ as the input and generates the prediction $\hat{\mathbf{Y}}$. The model is then optimized by encouraging the prediction to match the pre-defined target $\mathbf{Y}$:

$$\hat{\mathbf{Y}} = \mathcal{G}(\hat{\mathbf{I}}), \quad \mathcal{L} = \mathcal{D}(\mathbf{Y}, \hat{\mathbf{Y}}) \tag{1}$$

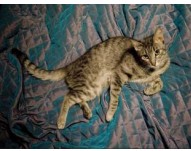 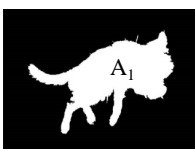 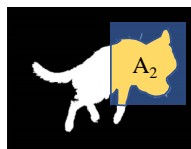

Figure 3: **Computation of object coverage percentage.** In the above example, $A_1$ is the foreground area. $A_2$ is the area of the yellow region. The blue rectangular is the cropped image produced by data augmentation. The object coverage percentage is obtained by the ratio between $A_2$ and $A_1$.

The loss $\mathcal{L}$ is computed according to a distance measurement $\mathcal{D}(\cdot)$ (*e.g.*, $L1$ or $L2$ distance) between the prediction and the target. In the following analysis, we investigate MAE (He et al., 2022) by diagnosing its reconstruction target and input image patches, identifying two important but previously overlooked bottlenecks that could have hurt the representation quality.

## 3.2 Empirical Analysis with MAE

**Reconstruction target.** MAE and most pixel-based MIM methods enforce the model to reconstruct intricate details of raw images. These complicated details contain textures with repeated patterns and belong to the high-frequency components in the frequency domain, which are usually independent of object shapes or scene structures. However, MAE tends to make significant efforts in encoding and reconstructing high-frequency details, as shown in Figure 1.

According to recent studies on shape and texture biases (Geirhos et al., 2021; 2019), vision models with stronger shape biases behave more like human visual perception, demonstrating better robustness and performing better when transferred to downstream tasks than those with stronger texture biases. Apparently, the current reconstruction target has introduced non-negligible texture biases, which deviate from the insights of previous works and might have hurt the representation quality. In Section 4.1, we provide a straightforward solution to de-emphasize the high-frequency components from the reconstruction target and justify its effectiveness with a quantitative analysis in Figure 5.

**Input patches.** To better understand the inputs to MIM methods at training time, we quantitatively measure how the input patches of MAE cover the foreground objects of raw images. Specifically, we adopt the binary object masks of ImageNet-1K generated by PSSL (Li et al., 2022b) and propose an object coverage percentage metric to evaluate an image processing operation $\mathcal{F}(\cdot)$, denoted by $\mathcal{J}(\mathcal{F})$. As illustrated by Figure 3, $\mathcal{J}(\mathcal{F})$ is defined as the ratio between areas $A_2$ and $A_1$. $A_1$ and $A_2$ are the areas of foreground objects in the original image $\mathbf{I}$ and the processed image $\mathcal{F}(\mathbf{I})$, respectively. We then leverage the metric to investigate how MAE's choice of $\mathcal{A}(\cdot)$ and $\mathcal{M}(\cdot)$ have influenced the object coverage. As discussed in Table 1, MAE employs the commonly used RRC for $\mathcal{A}(\cdot)$ and a masking operation with 75% mask ratio for $\mathcal{M}(\cdot)$. We found that $\mathcal{J}(\mathcal{A}) = 68.3\%$, but $\mathcal{J}(\mathcal{M} \circ \mathcal{A})$ sharply reduces to 17.1%, indicating a potential lack of foreground information in the inputs of MAE.

As argued by DeiT III (Touvron et al., 2022), the foreground usually encodes more semantics than the background, and a lack of foreground can result in sub-optimal optimization in supervised learning. In MIM, the coupling of RRC and aggressive masking might have hindered representation learning. In Section 4.2, we rigorously review various augmentation functions $\mathcal{A}(\cdot)$ and propose a simple workaround to preserve more foreground information in the input patches.

## 4 PixMIM

Based on the analysis, we develop a straightforward yet effective method, **PixMIM**, which addresses the two identified bottlenecks discussed in Section 3. PixMIM includes two strategies: 1) generating low-frequency reconstruction targets (Section 4.1), and 2) replacing the RRC with a more conservative augmentation (Section 4.2). An overview of PixMIM is presented in Figure 4.

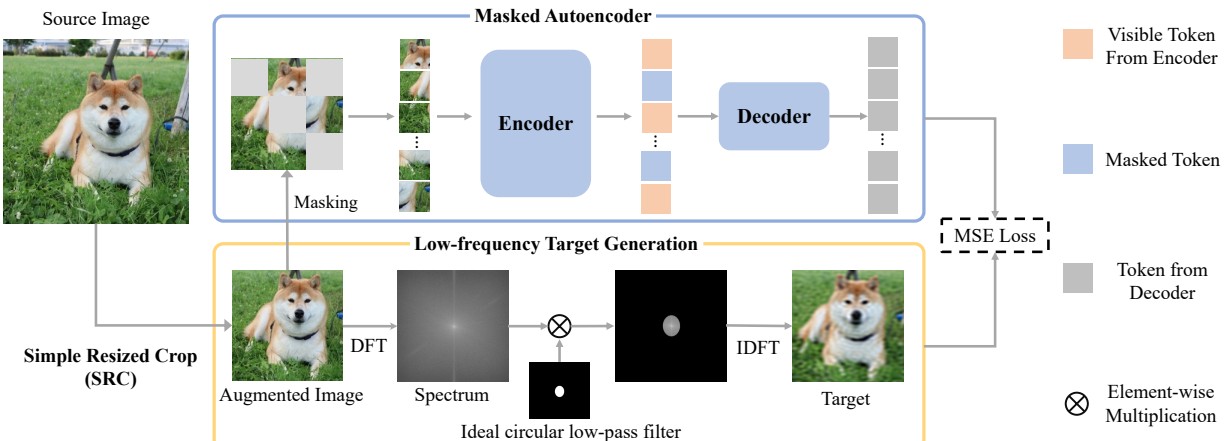

Figure 4: **The architecture of PixMIM.** Guided by the analysis (Section 3), PixMIM consists of straight-forward strategies: 1) generates low-frequency reconstruction targets to de-emphasize the texture-dominated details and prioritize the learning of low-frequency patterns (Section 4.1), and 2) replace the commonly used Random Resized Crop (RRC) with the less aggressive Simple Resized Crop (SRC) to alleviate the problem of missing foreground in the input patches (Section 4.2).

## 4.1 Low-frequency Target Generation

To de-emphasize the model from reconstructing texture-dominated high-frequency details, we propose a novel target generator $\mathcal{G}(\cdot)$, in which we maintain the target in RGB format for efficiency but filter out the high-frequency components. Specifically, we define the low-frequency target generation with the following three steps: 1) domain conversion from spatial to frequency, 2) low-frequency components extraction, and 3) reconstruction target generation from frequency domain (see Figure 4 for an illustration).

**Step-1: Domain conversion from spatial to frequency.** We use the one-channel image $\mathbf{I}_i \in \mathbb{R}^{H \times W}$ to demonstrate our approach for notation simplicity. With 2D Discrete Fourier Transform (DFT) $\mathcal{F}_{\text{DFT}}(\cdot)$, the frequency representation of the image could be derived by:

$$\mathcal{F}_{\text{DFT}}(\mathbf{I}_i)(u,v) = \sum_{h=0}^{H-1}\sum_{w=0}^{W-1}\mathbf{I}_i(h,w)e^{-i2\pi(\frac{uh}{H}+\frac{vw}{W})} \tag{2}$$

Where $(u,v)$ and $(h,w)$ are the frequency spectrum and spatial space coordinates, respectively. $\mathcal{F}_{\text{DFT}}(\mathbf{I}_i)(u,v)$ is the complex frequency value at $(u,v)$. $\mathbf{I}_i(h,w)$ is the pixel value at the $(h,w)$ and $i$ is the imaginary unit. Please refer to the Appendix for full details of the imaginary and real parts of $\mathcal{F}_{\text{DFT}}(\mathbf{I}_i)(u,v)$.

**Step-2: Low-frequency components extraction.** To only retain the low frequency components of the image $\mathbf{I}_i$, we apply an ideal low-pass filter $\mathcal{F}_{\text{LPF}}$ on the frequency spectrum $\mathcal{F}_{\text{DFT}}(\mathbf{I}_i)$. The ideal low-pass filter is defined as:

$$\mathcal{F}_{\text{LPF}}(u,v) = \begin{cases} 1, & \sqrt{((u-u_c)^2+(v-v_c)^2)} \leq r, \\ 0, & \text{otherwise.} \end{cases} \tag{3}$$

Where $v_c$ and $u_c$ are the center coordinates of the frequency spectrum. $r$ is the bandwidth of the circular ideal low-pass filter to control how many high-frequency components will be filtered out from the spectrum, and we have $r \in [0, \min(\frac{H}{2}, \frac{W}{2})]$. The extraction process is represented as $\mathcal{F}_{\text{LPF}}(u,v) \otimes \mathcal{F}_{\text{DFT}}(\mathbf{I}_i)(u,v)$, and $\otimes$ is the element-wise multiplication.

**Step-3: Reconstruction target generation.** We then apply the inverse Discrete Fourier Transform (IDFT) $\mathcal{F}_{\text{IDFT}}$ on the filtered spectrum to generate the RGB image as the final reconstruction target:

$$\mathbf{Y} = \mathcal{F}_{\text{IDFT}}(\mathcal{F}_{\text{LPF}}(u,v) \otimes \mathcal{F}_{\text{DFT}}(\mathbf{I}_i)(u,v)) \tag{4}$$

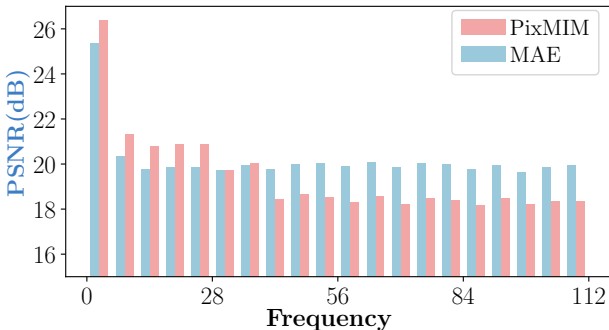

Figure 5: **Frequency analysis of MAE and PixMIM.** The PSNR of the reconstructed image for various frequency intervals (similar to Figure 1), averaged across 50,000 images from ImageNet-1K's validation set. PixMIM shifts the model's focus toward low-frequency components. Note that the figure illustrates the frequency response of images from the validation set rather than the training set, and some high-frequency components persist in the low-frequency region.

Both $\mathcal{F}_{\text{DFT}}$ and $\mathcal{F}_{\text{IDFT}}$ can be computed efficiently with Fast Fourier Transform (Brigham & Morrow, 1967). The computation cost of the above three steps is negligible thanks to the highly optimized implementation in PyTorch (Paszke et al., 2019).

To verify if our method successfully de-emphasizes the reconstruction of high-frequency components, Figure 5 presents a frequency analysis across 50,000 images from the validation set of ImageNet-1K, using $r = 40$. Compared to the vanilla MAE, our method produces obviously lower reconstruction PSNR at high-frequency intervals and slightly higher PSNR at low-frequency intervals, justifying the effectiveness of our method.

### 4.2 More Conservative Image Augmentation

Based on the analysis in Section 3, we would like to retain more foreground information in the input patches to our model. As a high masking ratio is crucial for MIM to learn effective representations (He et al., 2022), the most straightforward strategy is to keep the corruption $\mathcal{M}(\cdot)$ unchanged but make the augmentation function $\mathcal{A}(\cdot)$ more conservative.

We extend our quantitative analysis of object coverage to get Figure 6, which compares RRC with two less aggressive image augmentation operations. Simple Resized Crop (SRC) is the augmentation technique used in AlexNet (Krizhevsky et al., 2012), which resizes the image by matching the smaller edge to the pre-defined training resolution (*e.g.*, 224), then applies a reflect padding of 4 pixels on both sides, and finally randomly crops a square region of the specified training resolution to get the augmented image. Center Crop (CC) always takes the fixed-size crop from the center of the image. The results show that the SRC has much higher $\mathcal{J}(\mathcal{F})$ than RRC and CC. When the masking strategy of MAE is applied, SRC produces a $\mathcal{J}(\mathcal{F})$ of 22.1%, which is very close to the upper bound (*i.e.*, 25%). Therefore, we simply adopt the SRC as the augmentation function $\mathcal{A}(\cdot)$ and take the off-the-shelf implementation from (Touvron et al., 2022).

Note that when there is no image masking, the SRC raises the $\mathcal{J}(\mathcal{F})$ from the 68.3% of RRC to 88.2%, indicating that it offers less diversity than RRC, which accounts for the performance degeneration in supervised image classification observed by DeiT III (Touvron et al., 2022). But unlike supervised learning, the aggressive image masking in MIM already provides sufficient randomness, and the use of SRC will not hurt the diversity as in supervised learning.

### 4.3 Plug into existing MIM Methods

Unlike recent approaches such as CAE (Chen et al., 2022), MILAN (Hou et al., 2022), or BEiTv2 (Peng et al., 2022a), our method is lightweight and straightforward. It can easily be plugged into most existing pixel-based MIM frameworks. To demonstrate its effectiveness and versatility, we apply our method to

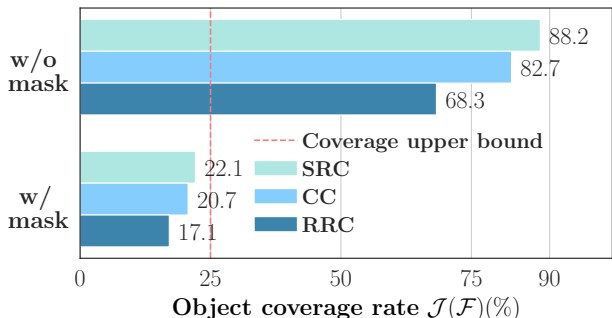

Figure 6: **Object coverage analysis.** SRC retains a higher proportion than RRC and CC, even under the aggressive masking strategy of MAE. Note that the upper bound of the object coverage is 25% when the masking strategy of MAE is applied. (RRC: random resized crop, SRC: simple resized crop, CC: center crop)

MAE (He et al., 2022), MFF (Liu et al., 2023), ConvMAE (Gao et al., 2022), and LSMAE (Hu et al., 2022) to obtain PixMIM$_{\text{MAE}}$, PixMIM$_{\text{MFF}}$, PixMIM$_{\text{ConvMAE}}$, and PixMIM$_{\text{LSMAE}}$ respectively. The experimental results are presented in the next section.

## 5 Experiments

In Section 5.1, we describe the experimental settings for pre-training and evaluation. Then in Section 5.2, we apply our method to four MIM baselines (*i.e.*, MAE (He et al., 2022), MFF (Liu et al., 2023), ConvMAE (Gao et al., 2022), and LSMAE (Hu et al., 2022)), compare the results with the state of the arts, and discuss the sensitivity of the ImageNet fine-tuning protocol. To complement the ImageNet fine-tuning protocol, Section 5.3 demonstrates additional analyses by checking the robustness of pre-trained models with out-of-distribution (OOD) ImageNet variants and conducting a shape bias analysis. Finally, Section 5.4 provides comprehensive ablation studies for our method.

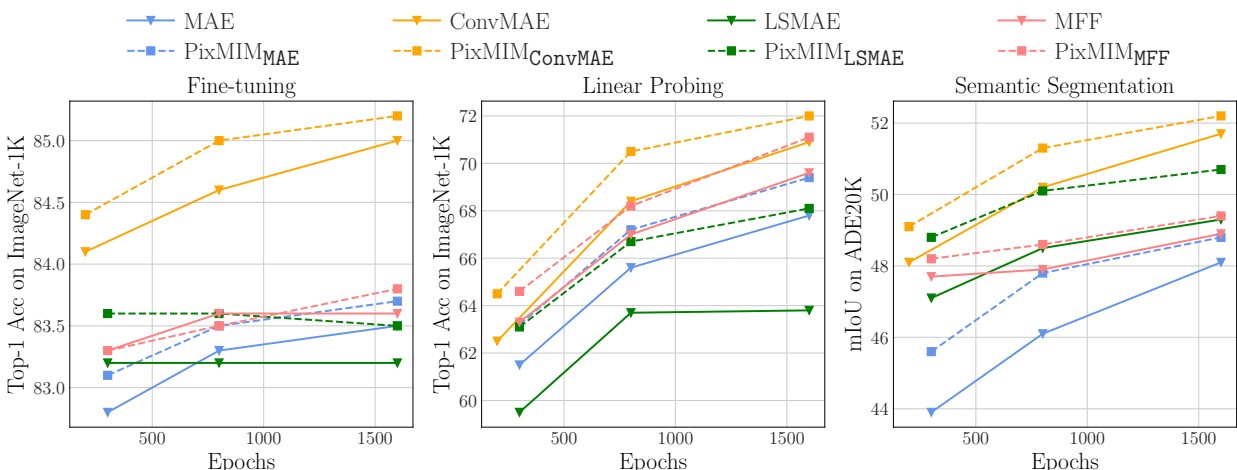

Figure 7: **Performance vs. epoch plots.** With different training epochs, PixMIM consistently brings significant gains to the baseline MIM approaches across various evaluation protocols.

### 5.1 Experiment Settings

We evaluate our methods and validate our design components with extensive experiments over image classification on ImageNet-1K (Deng et al., 2009), object detection on COCO (Lin et al., 2014), and semantic

| Evaluation Protocol→ | | | ImageNet | | COCO† | | ADE20K |
|---|---|---|---|---|---|---|---|
| Method | Target | Epoch | ft(%) | lin(%) | $AP^{box}$ | $AP^{mask}$ | mIoU |
| **Supervised learning** | | | | | | | |
| DeiT III (Touvron et al., 2022) | - | 800 | 83.8 | - | - | - | 49.3 |
| **Masked Image Modeling w/ pre-trained target generator** | | | | | | | |
| BEiT (Bao et al., 2021) | DALLE | 800 | 83.2 | 56.7 | - | - | 45.6 |
| CAE (Chen et al., 2022) | DALLE | 800 | 83.8 | 68.6 | 49.8 | 43.9 | 49.7 |
| MILAN (Hou et al., 2022) | CLIP-B | 400 | 85.4 | 78.9 | 52.6 | 45.5 | 52.7 |
| BEiT-v2 (Peng et al., 2022a) | VQ-KD | 1600 | 85.5 | 80.1 | - | - | 53.1 |
| MaskDistill (Peng et al., 2022b) | CLIP-B | 800 | 85.5 | - | - | - | 54.3 |
| **Masked Image Modeling w/o pre-trained target generator** | | | | | | | |
| MaskFeat (Wei et al., 2022) | HOG | 1600 | 84.0 | 62.3 | 52.3 | 46.4 | 48.3 |
| SemMAE (Li et al., 2022a) | RGB | 800 | 83.4 | 65.0 | - | - | 46.3 |
| SimMIM (Xie et al., 2022) | RGB | 800 | 83.8 | 56.7 | - | - | - |
| MAE* (He et al., 2022) | RGB | 800 | 83.3 | 65.6 | 51.3 | 45.7 | 46.1 |
| **PixMIM**$_{MAE}$ | RGB | 800 | 83.5 (+0.2) | 67.2 (+1.6) | 51.7 (+0.4) | 46.1 (+0.4) | 47.3 (+1.2) |
| MFF (Liu et al., 2023) | RGB | 800 | 83.6 | 67.0 | 51.8 | 46.1 | 47.9 |
| **PixMIM**$_{MFF}$ | RGB | 800 | 83.5 (-0.1) | 68.2 (+1.2) | 52.3 (+0.5) | 46.7 (+0.6) | 48.6 (+0.7) |
| ConvMAE* (Gao et al., 2022) | RGB | 800 | 84.6 | 68.4 | 52.0 | 46.3 | 50.2 |
| **PixMIM**$_{ConvMAE}$ | RGB | 800 | 85.0 (+0.4) | 70.5 (+2.1) | 53.1 (+1.1) | 47.0 (+0.7) | 51.3 (+1.1) |
| LSMAE* (Hu et al., 2022) | RGB | 800 | 83.2 | 63.7 | 51.0 | 45.4 | 48.5 |
| **PixMIM**$_{LSMAE}$ | RGB | 800 | 83.6 (+0.4) | 66.7 (+3.0) | 52.1 (+1.1) | 46.3 (+0.9) | 50.1 (+1.6) |

Table 2: **Performance comparison of MIM methods on various downstream tasks.** We report the results with fine-tuning (ft) and linear probing (lin) experiments on ImageNet-1K, objection detection on COCO, and semantic segmentation on ADE20K. The backbone of all experiments is ViT-B (Dosovitskiy et al., 2021). ∗: numbers are reported by running the official code release. †: As there is no uniform number of fine-tuning epochs for MAE, MFF, ConvMAE, and LSMAE for object detection, we fine-tuned PixMIM using the same number of epochs as each respective base method.

segmentation on ADE20K (Zhou et al., 2018). Unless otherwise specified, we report the performance with ViT-B (Dosovitskiy et al., 2021).

**ImageNet-1K (Deng et al., 2009)**   ImageNet-1K consists of 1.3M images of 1k categories and is split into the training and validation sets. When applying our methods to MAE (He et al., 2022), MFF (Liu et al., 2023), ConvMAE (Gao et al., 2022), and LSMAE (Hu et al., 2022), we strictly follow their original pre-training and evaluation settings on ImageNet-1K to guarantee the fairness of experiments, including the pre-training schedule, network architecture, learning rate setup, and fine-tuning protocols, etc. The only exception is that we increase the batch size of ConvMAE from 1024 to 4096 to accelerate the pre-training, while this change does not affect the performance according to our observations. We provide complete implementation details in the Appendix.

**ADE20K (Zhou et al., 2018)**   For the semantic segmentation experiments on ADE20K, we follow the basic off-the-shelf settings from MAE (He et al., 2022). A UperNet (Xiao et al., 2018) is fine-tuned for 160k iterations with a batch size of 16. In addition, we also turn on the relative position bias and initialize them with zero. We report the Mean Intersection over Union (mIoU) results averaged over two runs for a robust comparison. The full details can be found in the Appendix.

**COCO (Lin et al., 2014)**   For object detection experiments on COCO, we adopt the Mask R-CNN approach (He et al., 2017) that produces bounding boxes and instance masks simultaneously, with the ViT as the backbone. Similar to MAE, we employ the box and mask AP as the metrics. For MAE and LSMAE, we use the official implementation of ViTDet (Li et al., 2022c). For ConvMAE, we use its released official repository. More detailed settings can be found in the Appendix.

| Method | Target | Bakcbone | ft | lin | seg |
|---|---|---|---|---|---|
| LSMAE (Hu et al., 2022) | RGB | ViT-B | 83.2 | 63.7 | 48.5 |
| MAE (He et al., 2022) | RGB | ViT-B | 83.3 | 65.6 | 46.1 |
| PixMIM$_{\text{MAE}}$ | RGB | ViT-B | 83.5 | 67.2 | 47.8 |
| PixMIM$_{\text{MFF}}$ | RGB | ViT-B | 83.5 | 68.2 | 48.6 |
| PixMIM$_{\text{LSMAE}}$ | RGB | ViT-B | 83.6 | 66.7 | 50.1 |
| SimMIM (Xie et al., 2022) | RGB | ViT-B | 83.8 | 56.7 | - |
| MaskFeat (Wei et al., 2022) | HOG | ViT-B | 84.0 | 62.3 | 48.3 |

Table 3: **Investigating the ImageNet fine-tuning protocol.** Six MIM approaches are *sorted* based on their ImageNet fine-tuning (ft) performance. The fine-tuning result alone hardly distinguishes different approaches with the same backbone and not using an extra pre-trained model for generating training targets, and it does not necessarily correlate with other evaluation protocols. Best viewed in color.

**Ablation studies** All ablation studies are based on the MAE settings. Following the common practice of previous MIM works (Liu et al., 2022a; Chen et al., 2022; Gao et al., 2022), we pre-train all model variants on ImageNet-1K for 300 epochs and comprehensively compare their performance on linear probing, fine-tuning, and semantic segmentation. All other settings are the same as those discussed above.

## 5.2 Main Results

In Table 2, we show the results of applying our simple method to MAE (He et al., 2022), MFF Liu et al. (2023), ConvMAE (Gao et al., 2022), and LSMAE (Hu et al., 2022), and compare these results with the state-of-the-art MIM approaches. Without extra computational cost, we consistently improve the original MAE, ConvMAE, MFF, and LSMAE across all downstream tasks. The margins on linear probing, object detection, and semantic segmentation are remarkable. Specifically, PixMIM$_{\text{LSMAE}}$ significantly improves the original LSMAE on linear probing and semantic segmentation by 3.0% and 1.6%, respectively. To further demonstrate the effectiveness of our method across various pre-training schedules, we plot the *performance vs. epoch* curves in Figure 7. The curves of PixMIM$_{\text{MAE}}$, PixMIM$_{\text{ConvMAE}}$, PixMIM$_{\text{MFF}}$, and PixMIM$_{\text{LSMAE}}$ consistently remain above the corresponding base methods by clear gaps. All these results demonstrate the universality and scalability of our methods. Additionally, we present the outcomes of applying PixMIM to backbones of varying scales, such as ViT-S and ViT-L. Please refer to Table 16 for further details.

**Methods with pre-trained target generator.** Although the methods with a powerful pre-trained target generator (Hou et al., 2022; Peng et al., 2022a) achieve the best results in Table 2, they rely on extra pre-training data and bring significant computational overhead to MIM when generating targets dynamically. In contrast, our improvements come with negligible cost and take a step towards closing the gap between pixel-based approaches and those relying on pre-trained target generators.

**Remarks on the ImageNet fine-tuning protocol.** According to Table 2, the improvements brought by our method on the ImageNet fine-tuning protocol are less obvious than those on the other three protocols. Table 3 investigates the correlation between the evaluation protocols by *sorting* six MIM approaches based on the ImageNet-finetuning performance, and we have the following observations:

- With the same network backbone and not using extra pre-trained models for generating training targets, the ImageNet fine-tuning performances of various methods always show marginal gaps.
- Better result on ImageNet fine-tuning does not necessarily mean better performance on linear probing or semantic segmentation. This is also shown by the curves of LSMAE in Figure 7.

Hence, we argue that ImageNet fine-tuning is *not a sensitive metric*, and we should include more protocols for comprehensively evaluating the representation quality. A potential explanation provided by CAE (Chen et al., 2022) is that the pre-training and fine-tuning data follow the same distribution and can narrow the gap among different methods. We provide additional analyses in the next subsection to complement the ImageNet fine-tuning protocol.

| Method | IN-C↓ | IN-A | IN-R | IN-S |
|---|---|---|---|---|
| LSMAE | 48.8 | 34.2 | 50.3 | 36.2 |
| PixMIM$_{\text{LSMAE}}$ | 48.0 (-0.8) | 36.1 (+1.9) | 50.8 (+0.5) | 37.1 (+0.9) |
| MAE | 51.7 | 35.9 | 48.3 | 34.5 |
| PixMIM$_{\text{MAE}}$ | 49.9 (-1.8) | 37.1 (+1.2) | 49.6 (+1.3) | 35.9 (+1.4) |
| ConvMAE | 45.5 | 50.8 | 54.6 | 41.1 |
| PixMIM$_{\text{ConvMAE}}$ | 45.3 (-0.2) | 52.5 (+1.7) | 55.3 (+0.7) | 41.8 (+0.7) |
| MFF | 49.0 | 37.2 | 51.0 | 36.8 |
| PixMIM$_{\text{MFF}}$ | 48.5 (-0.5) | 40.1 (+2.9) | 51.6 (+0.6) | 37.8 (+1.0) |

Table 4: **Robustness evaluation on ImageNet variants.** To complement the less sensitive ImageNet fine-tuning protocol, we further evaluate the fine-tuned models from the main table on four ImageNet variants. Results are reported in top-1 accuracy, except for IN-C which uses the mean corruption error.

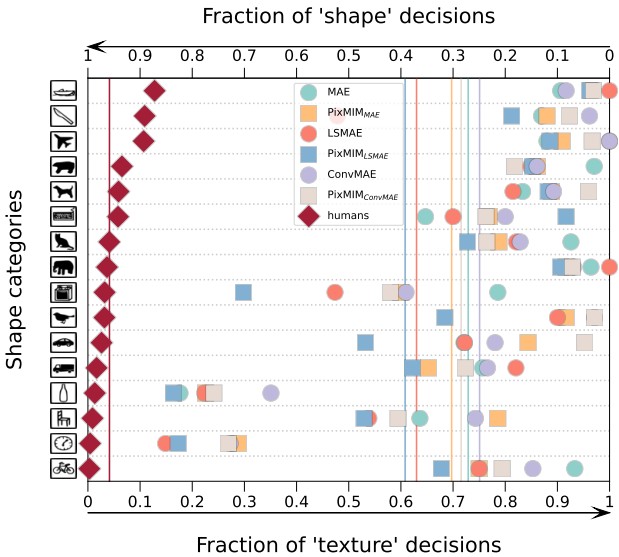

Figure 8: **Shape bias analysis.** PixMIM consistently improves the shape bias of the baselines. Each vertical line is the *weighted average* of all 16 categories.

## 5.3 Additional Analyses

Two additional experiments are presented to complement the less sensitive ImageNet fine-tuning protocol and further validate the effectiveness of our method.

**Robustness checking.** we compare the pre-trained models on four out-of-distribution ImageNet variants: ImageNet-Corruption (Hendrycks & Dietterich, 2019), ImageNet-Adversarial (Hendrycks et al., 2021b), ImageNet-Rendition (Hendrycks et al., 2021a), and ImageNet-Sketch (Wang et al., 2019). These datasets introduce various domain shifts to the original ImageNet-1K and are widely used to assess a model's robustness and generalization ability. Table 4 shows that PixMIM$_{\text{MAE}}$, PixMIM$_{\text{ConvMAE}}$, PixMIM$_{\text{LSMAE}}$, and PixMIM$_{\text{MFF}}$ consistently outperform their baselines, and the margins of improvement are much more pronounced than those on the validation set of Imagenet-1K. The better robustness against domain shifts strengthens the value of our simple yet effective method.

**Shape bias analysis.** We take the off-the-shelf shape bias toolbox (Geirhos et al., 2021) to analyze our pre-trained models. Shape bias measures how much the model relies on shapes to extract the semantic representation of the image, quantified as the fraction of correct decisions based on object shape. Figure 8 shows that PixMIM$_{\text{MAE}}$, PixMIM$_{\text{ConvMAE}}$, and PixMIM$_{\text{LSMAE}}$ improve the shape bias of their baselines, confirming

that our methods prevent the model from being excessively texture-biased by filtering out the high-frequency components of the target image. The colored lines denote the *weighted average* of shape bias across different categories for different methods.

| $r$ | ft | lin | seg |
|---|---|---|---|
| baseline | 82.8 | 61.5 | 43.9 |
| 30 | 82.9 | 62.5 | 44.6 |
| 35 | **82.9** | 62.6 | 44.8 |
| 40 | 82.8 | **62.7** | **45.3** |
| 45 | 82.7 | 62.2 | 45.1 |
| 50 | 82.8 | 61.8 | 44.8 |

(a) The bandwidth($r$) of the low-pass filter. $r = 40$ yields the optimal result.

| aug | ft | lin | seg |
|---|---|---|---|
| RRC | 82.8 | 61.5 | 43.9 |
| SRC | **82.9** | **62.5** | **44.3** |
| CC | 82.8 | 62.2 | 44.1 |
| BG | 82.5 | 59.5 | 43.4 |
| Resize | 82.6 | 58.2 | 42.1 |

(b) The data augmentation. BG: background-focused random cropping, see text for details. Resize: resize the image, no random cropping.

| LF | SRC | ft | lin | seg |
|---|---|---|---|---|
| - | - | 82.8 | 61.5 | 43.9 |
| ✓ | - | 82.8 | 62.7 | 45.3 |
| - | ✓ | 82.9 | 62.5 | 44.3 |
| ✓ | ✓ | **83.2** | **63.3** | **46.2** |

(c) The combination of the low-frequency target (LF) and simple resized crop (SRC).

Table 5: **The ablation studies** with ViT-B/16 pre-trained on ImageNet-1K for 300 epochs. We report fine-tuning (ft), linear probing (lin), and semantic segmentation (seg) results. Our final settings are marked in  gray .

### 5.4 Ablation Studies

We further conduct ablation studies for our key design components: the filtering of the high-frequency components of the target image and the use of the Simple Resized Crop.

**The bandwidth of the low-pass filter.** Table 5a investigates how varying the bandwidth $r$ influences the vanilla MAE. All model variants in the table are trained for 300 epochs following the training recipes of MAE. The optimal bandwidth is 40, and it improves the baseline significantly (*i.e.*, +1.2% on linear probing and +1.7% on semantic segmentation). A narrow bandwidth could discard important information about the image (*e.g.* edges of objects), leading to a performance drop. In comparison, a too-large bandwidth fails to remove unessential textures effectively.

**Replace RRC with SRC.** Table 5b compares different data augmentations. The simple resized crop (SRC) brings non-trivial improvement to the original random resized crop (RRC) used by MAE on both linear probing and semantic segmentation. However, recall in DeiT III (Touvron et al., 2022), replacing RRC with SRC degrades the performance as it decreases the cropped image's diversity and impairs the model's generalization ability. The opposite results we obtain in MIM here suggest that the RRC could have led to the severe issue of missing foreground, which is further confirmed by the fact that even the simple center crop can outperform RRC in linear probing and semantic segmentation.

To better support our analysis on the input patches of MAE, we conduct reverse engineering of SRC, which crops mostly the background region instead of the foreground (*i.e.*, the BG entry in Table 5b). Please check the Appendix for implementation details and visualizations. The results demonstrate that the absence of foreground information can significantly impair the representation quality, further confirming our analysis regarding the input patches.

Table 5c further verifies that the gains brought by the two components of PixMIM effectively accumulate over all three evaluation protocols. We also extend Table 5b to a non-object-centric dataset, Place365 (López-Cifuentes et al., 2020). Table 6 shows that SRC still brings non-negligible improvements over RRC when pre-trained on scene-scale images, suggesting that the lacking foreground issue in pixel-based MIM is universal and not specific to object-centric datasets (*e.g.*, ImageNet).

| Method | RRC | SRC | ft | lin | seg |
|--------|-----|-----|-----|-----|-----|
| MAE | ✓ | | 82.1 | 48.4 | 45.8 |
| MAE | | ✓ | 82.4 (+0.3) | 48.9 (+0.5) | 46.8 (+1.0) |

Table 6: Extension of Table 5b, pre-trained on the non-object-centric dataset Place365 (López-Cifuentes et al., 2020). The lacking foreground issue is not unique to MIM training on single-object datasets like ImageNet.

## 6 Limitation and Future Works

Currently, our experiments are based on ViT-B (Dosovitskiy et al., 2021), which is also the common practice by some other works (Xie et al., 2022; Li et al., 2022a). Some studies, like (He et al., 2022), suggest that after extending experiments to larger models, *e.g.*, ViT-L or ViT-H, the same method can still get equivalent gains, but evaluating the scalability of our methods on larger models is also expected. In addition, the bandwidth $r$ is designed as a hyper-parameter and may vary across different datasets or input resolutions. So a self-adaptive bandwidth is also expected. Finally, self-supervised pre-training has been criticized for consuming many computational resources. Even though our method brings negligible computation overhead, making the entire pre-training pipeline more efficient should be one of the directions for future research.

## 7 Conclusion

In this paper, we first provide an empirical analysis of the milestone algorithm, MAE, from the perspective of input patches and reconstruction targets, identifying the potential bottlenecks of existing pixel-based MIM approaches. Based on the analysis, we propose a simple yet effective method, PixMIM, without introducing extra computation overhead or complicating the pre-training pipeline. When applied to three representative pixel-based MIM approaches, PixMIM brings consistent performance boosts across various downstream tasks and improves the model's robustness, demonstrating its effectiveness and universality.

## 8 Acknowledgement

This paper is supported by the National Key R&D Program of China (No. 2022ZD0161600) and Shanghai Postdoctoral Excellence Program (No.2022235).

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

# Appendix

## A  Discrete Fourier Transform

$\mathcal{F}_{\text{DFT}}(\mathbf{I}_i)(u, v)$ in Section 4.1 of the main paper can be decomposed into real and imaginary parts:

$$\mathcal{F}_{\text{DFT}}(\mathbf{I}_i)(u, v) = \mathcal{R}(\mathrm{I}_i)(u, v) + \mathcal{I}(\mathrm{I}_i)(u, v)i \tag{5}$$

Both $\mathcal{R}(\mathrm{I}_i)(u, v)$ and $\mathcal{I}(\mathrm{I}_i)(u, v)$ are real numbers, and $i$ is the imaginary unit. The amplitude and phase of $\mathcal{F}_{\text{DFT}}(\mathbf{I}_i)(u, v)$ can be obtained using the following formulas:

$$\mathcal{A}(\mathrm{I}_i)(u, v) = (\mathcal{R}(\mathrm{I}_i)(u, v)^2 + \mathcal{I}(\mathrm{I}_i)(u, v)^2)^{\frac{1}{2}} \tag{6}$$

$$\mathcal{P}(\mathrm{I}_i)(u, v) = arctan(\frac{\mathcal{R}(\mathrm{I}_i)(u, v)}{\mathcal{I}(\mathrm{I}_i)(u, v)}) \tag{7}$$

## B  Full Implementation Details

### B.1  Pre-training

We show the detailed implementation settings for the pre-training of PixMIM$_{\texttt{MAE}}$, PixMIM$_{\texttt{ConvMAE}}$, PixMIM$_{\texttt{MFF}}$ and PixMIM$_{\texttt{LSMAE}}$ in the following tables. While for the result for LSMAE (Hu et al., 2022) in Table 2 of the main paper, except for that we decrease the batch size from 4096 to 2048 due to the limited computational resources, other settings are kept the same.

| config | value |
|---|---|
| optimizer | AdamW  (Loshchilov & Hutter, 2019) |
| base learning rate | 1.5e-4 |
| weight decay | 0.05 |
| optimizer momentum | $\beta_1, \beta_2$=0.9, 0.95  (Chen et al., 2020a) |
| batch size | 4096 |
| learning rate schedule | cosine decay  (Loshchilov & Hutter, 2017) |
| warmup epochs  (Goyal et al., 2017) | 40 |
| augmentation | SimpleResizedCrop |

Table 7: Pre-training setting of PixMIM$_{\texttt{MAE}}$, PixMIM$_{\texttt{MFF}}$, and PixMIM$_{\texttt{ConvMAE}}$.

| config | value |
|---|---|
| optimizer | AdamW  (Loshchilov & Hutter, 2019) |
| base learning rate | 1.5e-4 |
| weight decay | 0.05 |
| optimizer momentum | $\beta_1, \beta_2$=0.9, 0.95  (Chen et al., 2020a) |
| batch size | 2048 |
| learning rate schedule | cosine decay  (Loshchilov & Hutter, 2017) |
| warmup epochs (Goyal et al., 2017) | 40 |
| augmentation | SimpleResizedCrop |

Table 8: Pre-training setting of PixMIM$_{\texttt{LSMAE}}$.

### B.2  ImageNet Fine-tuning

The implementation details for the fine-tuning of PixMIM$_{\texttt{MAE}}$, PixMIM$_{\texttt{ConvMAE}}$, and PixMIM$_{\texttt{LSMAE}}$ are shown in Table 9, which strictly follow that of MAE (He et al., 2022).

| config | value |
|---|---|
| optimizer | AdamW (Loshchilov & Hutter, 2019) |
| base learning rate | 1e-3 |
| weight decay | 0.05 |
| optimizer momentum | $\beta_1, \beta_2 = 0.9, 0.999$ |
| layer-wise lr decay (Clark et al., 2020; Bao et al., 2021) | 0.75 |
| batch size | 1024 |
| learning rate schedule | cosine decay |
| warmup epochs | 5 |
| training epochs | 100 |
| augmentation | RandAug (9, 0.5) (Cubuk et al., 2020) |
| label smoothing (Szegedy et al., 2016) | 0.1 |
| mixup (Zhang et al., 2018) | 0.8 |
| cutmix (Yun et al., 2019) | 1.0 |
| drop path (Huang et al., 2016) | 0.1 |

Table 9: End-to-end fine-tuning setting of PixMIM$_{\texttt{MAE}}$, PixMIM$_{\texttt{ConvMAE}}$, PixMIM$_{\texttt{MFF}}$, and PixMIM$_{\texttt{LSMAE}}$.

### B.3 ImageNet Linear Probing

The implementation details for PixMIM$_{\texttt{MAE}}$ and PixMIM$_{\texttt{LSMAE}}$ are shown in Table 10, which also follow that of MAE (He et al., 2022). For PixMIM$_{\texttt{ConvMAE}}$, we decrease the batch size from 16384 to 4096, following the official setting.

| config | value |
|---|---|
| optimizer | LARS (You et al., 2017) |
| base learning rate | 0.1 |
| weight decay | 0 |
| optimizer momentum | 0.9 |
| batch size | 16384 |
| learning rate schedule | cosine decay |
| warmup epochs | 10 |
| training epochs | 90 |
| augmentation | RandomResizedCrop |

Table 10: Linear probing setting of PixMIM$_{\texttt{MAE}}$, PixMIM$_{\texttt{MFF}}$, and PixMIM$_{\texttt{LSMAE}}$.

| config | value |
|---|---|
| optimizer | LARS (You et al., 2017) |
| base learning rate | 0.1 |
| weight decay | 0 |
| optimizer momentum | 0.9 |
| batch size | 4096 |
| learning rate schedule | cosine decay |
| warmup epochs | 10 |
| training epochs | 90 |
| augmentation | RandomResizedCrop |

Table 11: Linear probing setting of PixMIM$_{\texttt{ConvMAE}}$.

### B.4 ADE20K Semantic Segmentation

The implementation details for PixMIM$_{\texttt{MAE}}$ and PixMIM$_{\texttt{LSMAE}}$ are shown in Table 12, and those for PixMIM$_{\texttt{ConvMAE}}$ are in Table 13.

| config | value |
|---|---|
| optimizer | AdamW (Loshchilov & Hutter, 2019) |
| base learning rate | 2e-4 |
| weight decay | 0.05 |
| optimizer momentum | $\beta_1, \beta_2 = 0.9, 0.999$ |
| layer-wise lr decay (Clark et al., 2020; Bao et al., 2021) | 0.75 |
| batch size | 16 |
| learning rate schedule | cosine decay |
| warmup iters | 1500 |
| training iters | 160000 |
| drop path (Huang et al., 2016) | 0.1 |

Table 12: End-to-end semantic segmentation setting of PixMIM$_{\texttt{MAE}}$, PixMIM$_{\texttt{MFF}}$, and PixMIM$_{\texttt{LSMAE}}$.

| config | value |
|---|---|
| optimizer | AdamW (Loshchilov & Hutter, 2019) |
| base learning rate | 3e-4 |
| weight decay | 0.05 |
| optimizer momentum | $\beta_1, \beta_2 = 0.9, 0.999$ |
| layer-wise lr decay (Clark et al., 2020; Bao et al., 2021) | 0.75 |
| batch size | 16 |
| learning rate schedule | cosine decay |
| warmup iters | 1500 |
| training iters | 160000 |
| drop path (Huang et al., 2016) | 0.1 |

Table 13: End-to-end semantic segmentation setting of PixMIM$_{\texttt{ConvMAE}}$.

## B.5 COCO Object Detection

For PixMIM$_{\texttt{MAE}}$, we follow the official release in Detectron2 (Wu et al., 2019), which fine-tunes the pre-trained model end to end for 100 epochs. While for PixMIM$_{\texttt{LSMAE}}$ and PixMIM$_{\texttt{ConvMAE}}$, we follow the official pre-training epochs, which are 50 and 25 epochs, respectively. Other key hyper-parameters are in Table 14 and Table 15.

## C Additional Results

**Pre-training for 1600 epochs.** Table 17 provides the results of pre-training PixMIM for 1600 epochs and compares it against the corresponding base methods. PixMIM can still bring non-trivial improvements over base methods on various downstream tasks, which verifies the scalability of our method across pre-training epochs.

**Pre-training with other model variants** In this section, we provide addition results about ViT-S and ViT-L, and pre-train the models fro 1600 epochs. The detailed results are shown in Table 16, and PixMIM still bring consistent improvement over other model variants.

**The 2× object detection protocol.** We also provide the results with the 2× settings of object detection for PixMIM$_{\texttt{MAE}}$ in Table 18.

**Few-shot fine-tuning.** Table 19 presents the results of few-shot fine-tuning, which uses 1% and 10% of ImageNet-1K training data to pre-train the model.

**Pre-train MAE with background-focused image crops.** We provide the implementation details for the background-focused random cropping in Table 5b of the main paper. To crop the background, we use the ImageNet-1K binary mask from PSSL (Li et al., 2022b) in the RRC (implementedy by PyTorch (Paszke et al., 2019)) to check whether the cropped image contains less than 20% of the object in the original image. If the condition is satisfied, we return the cropped image. Otherwise, we crop another region of the image.

| config | value |
|---|---|
| optimizer | AdamW (Loshchilov & Hutter, 2019) |
| base learning rate | 8e-5 |
| weight decay | 0.1 |
| optimizer momentum | $\beta_1, \beta_2$=0.9, 0.999 |
| layer-wise lr decay (Clark et al., 2020; Bao et al., 2021) | 0.75 |
| batch size | 64 |
| learning rate schedule | cosine decay |
| training epochs | 100/50 |
| drop path (Huang et al., 2016) | 0.1 |

Table 14: End-to-end object detection setting of PixMIM$_{\texttt{MAE}}$, PixMIM$_{\texttt{MFF}}$, and PixMIM$_{\texttt{LSMAE}}$.

| config | value |
|---|---|
| optimizer | AdamW (Loshchilov & Hutter, 2019) |
| base learning rate | 1e-4 |
| weight decay | 0.1 |
| optimizer momentum | $\beta_1, \beta_2$=0.9, 0.999 |
| layer-wise lr decay (Clark et al., 2020; Bao et al., 2021) | 0.85 |
| batch size | 16 |
| learning rate schedule | cosine decay |
| training epochs | 25 |
| drop path (Huang et al., 2016) | 0.1 |

Table 15: End-to-end object detection setting of PixMIM$_{\texttt{ConvMAE}}$.

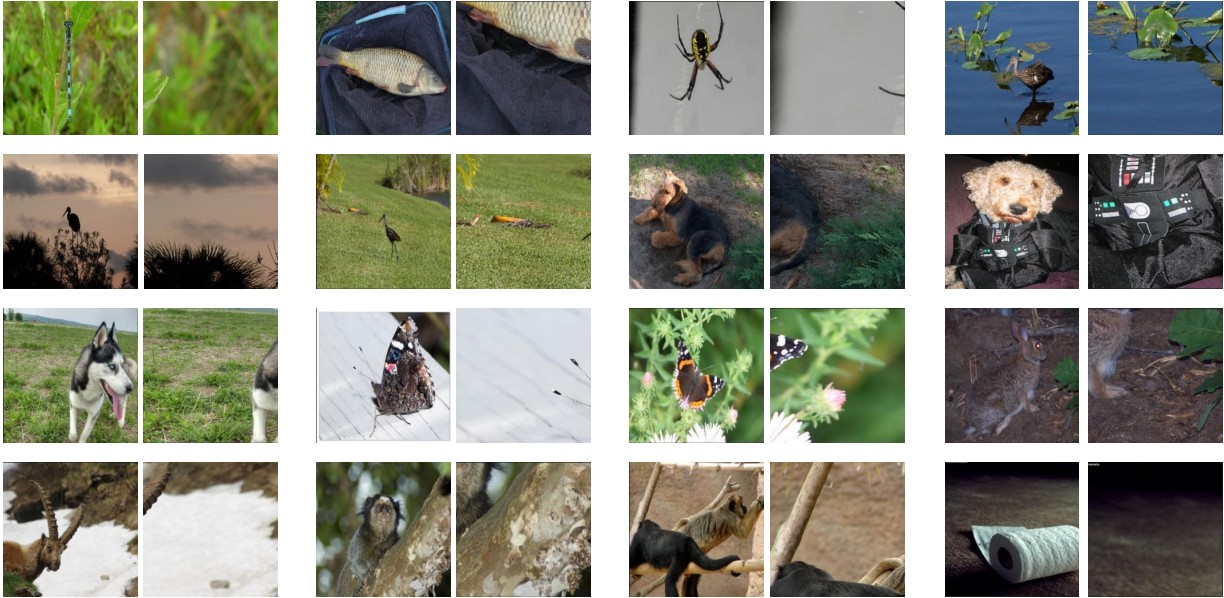

Figure 9: **Pre-train MAE with background cropped images.** Instead of cropping the foreground region in each image, we attempt to input the background into the model. The original image is depicted on the left of each pair, while the cropped background can be seen on the right.

We repeat the above procedure 50 times until the condition is satisfied; otherwise, we use the original RRC. Some visualizations of this augmentation strategy can be viewed in Figure 9.

**Applying PixMIM on methods using high-semantic features as targets.** The primary objective of PixMIM is to address the inherent bias of pixel-based MIM approaches, which tend to excessively prioritize low-level details. As a result, PixMIM has limited impact on methods that utilize high-semantic features as

| Method | Epoch | Model | LIN | SEG | FT |
|---|---|---|---|---|---|
| MAE | 1600 | ViT-S | 51.1 | 42.3 | 79.5 |
| PixMIM$_\text{MAE}$ | 1600 | ViT-S | 53.2(+2.1) | 44.1(+1.9) | 80.4(0.9) |
| MAE | 1600 | ViT-L | 76.0 | 53.6 | 85.9 |
| PixMIM$_\text{MAE}$ | 1600 | ViT-L | 77.1(+1.1) | 54.4(+0.8) | 86.1(+0.2) |

Table 16: Performance of applying PixMIM to other model variants.

| Method | Epoch | LIN | SEG | DET | FT |
|---|---|---|---|---|---|
| MAE | 1600 | 67.8 | 48.1 | 51.4 | 83.5 |
| PixMIM$_\text{MAE}$ | 1600 | 69.3(+1.5) | 48.7(+0.6) | 52.1(+0.7) | 83.6 |
| ConvMAE | 1600 | 70.9 | 51.7 | 53.6 | 85.0 |
| PixMIM$_\text{ConvMAE}$ | 1600 | 72.0(+1.1) | 52.2(+0.5) | 53.8(+0.2) | 85.2 |
| MFF | 1600 | 69.6 | 48.9 | 52.3 | 83.6 |
| PixMIM$_\text{MFF}$ | 1600 | 71.1(+1.5) | 49.4(+0.5) | 52.7(+0.4) | 83.8 |
| LSMAE | 1600 | 63.8 | 49.3 | 51.6 | 83.2 |
| PixMIM$_\text{LSMAE}$ | 1600 | 68.1(+4.3) | 50.7(+1.4) | 53.0(+1.4) | 83.5 |

Table 17: Performance Comparison by pre-training models for 1600 epochs.

| Method | Epoch | AP$^\text{box}$ | AP$^\text{mask}$ |
|---|---|---|---|
| MAE | 800 | 47.3 | 42.3 |
| PixMIM$_\text{MAE}$ | 800 | 47.8 (+0.5) | 42.8 (+0.5) |
| MAE | 1600 | 48.6 | 43.5 |
| PixMIM$_\text{MAE}$ | 1600 | 49.3 (+0.7) | 44.0 (+0.5) |

Table 18: Results of 2× settings for object detection.

| Method | Epoch | 1% | 10% |
|---|---|---|---|
| MAE | 800 | 45.4 | 71.2 |
| PixMIM$_\textbf{MAE}$ | 800 | 47.9 (+2.5) | 72.2 (+1.0) |

Table 19: Few-shot fine-tuning with 1% and 10% of ImageNet-1K training data.

targets, such as MILAN, MaskDistill, and EVA. To validate this assertion, we applied PixMIM to MILAN, and the results are presented in Table 20.

**Using an adaptive filtering threshold.** Rather than manually determining an optimal threshold through a hand-crafted ablation study, we also tried an adaptive filtering threshold (r) that can dynamically adapt within the range of 20 to 60 as a trainable parameter. As presented in Table 21, employing such a dynamic filtering threshold yields comparable results to using the current manually selected threshold.

**Suppressing the low-frequency components.** The low-frequency components are highly semantic, playing a crucial role in understanding the key information present in the image Bao et al. (2021); Ramesh et al. (2021); Rombach et al. (2022). Removing this type of information can have catastrophic consequences on the model's performance. To verify this, we conducted an additional ablation experiment where we removed all components below r=40. The results are presented in Table 22. As depicted in the table, the model's performance significantly deteriorates upon the removal of these low-frequency components from the image.

| Method | Epoch | Model | LIN | SEG | FT |
|---|---|---|---|---|---|
| MILAN | 400 | ViT-B | 79.9 | 52.7 | 85.4 |
| PixMIMMILAN | 400 | ViT-B | 79.7 | 52.7 | 85.5 |

Table 20: Performance of applying PixMIM to MILAN.

| Method | Epoch | Model | LIN | SEG | FT |
|---|---|---|---|---|---|
| Fix(r=40) | 800 | ViT-B | 65.6 | 46.1 | 83.3 |
| Adaptive | 800 | ViT-B | 65.6 | 46.3 | 83.4 |

Table 21: Using an adaptive filtering threshold yields similar performance to the current manually selected threshold.

| Method | Epoch | Model | LIN | SEG | FT |
|---|---|---|---|---|---|
| MAE | 800 | ViT-B | 65.6 | 46.1 | 83.3 |
| Removing Low-freq | 800 | ViT-B | 12.1 | 8.6 | 30.2 |

Table 22: Suppressing the low-frequency components in the training images can significantly hurt the performance of MAE.

