# OpenReview forum: "PixMIM: Rethinking Pixel Reconstruction in Masked Image Modeling"
_TMLR — Accepted by TMLR_

### Review · Reviewer_y8Tv · 2023-10-28

**Summary Of Contributions:**

The paper proposes two strategies, Low-frequency target generation and SRC data augmentation, to improve the performance of RGB-based MIM methods. The paper provides extensive analysis to show the limitations of previous methods and the effectiveness of the proposed solution. The solution is simple and compatible with existing RGB-based MIM methods without extra training costs.

**Audience:**

Yes

**Broader Impact Concerns:**

No concerns on the ethical implications.

**Claims And Evidence:**

Yes

**Requested Changes:**

Additional analysis on ViT-L or ViT-H baselines.
Better designs for low-frequency strategy.

**Strengths And Weaknesses:**

Strengths:
- The paper gives a clear and intuitive explanation of the proposed Low-frequency target generation and SRC strategies.
- The paper is well-written and easy to follow. The solution does not require extra training cost or complex modifications.
- The paper conducts thorough analysis to evaluate the impact of SRC from different perspectives. The reverse engineering of SRC demonstrates its benefits to MAE. The experiment on Place365 dataset shows the generalization of SRC to non-object-centric scenarios.

Weaknesses:
- The paper mainly applies existing techniques, such as low-frequency and SRC, to address the training target issue in RGB-based MIM methods. The novelty of the paper lies in the analysis rather than a novel solution. The paper could provide more novel and comprehensive designs and evaluations. In particular,
1) The paper sets the bandwidth r as a hyper-parameter that may vary across different datasets or input resolutions. It would be better to design a self-adaptive bandwidth that can adjust to different scenarios;
2) The paper only evaluates the solution on ViT-S and ViT-B baselines, which have relatively small capacity. It would be interesting to see how the solution performs on ViT-L or ViT-H baselines, which may be less sensitive to high-frequency details.
- The paper does not apply SRC to pre-trained teacher based methods, such as MILAN and MaskDistill, which could also benefit from the data augmentation strategy. The paper could compare the proposed solution with these methods to show its advantages.

---

### Review · Reviewer_khNT · 2023-11-03

**Summary Of Contributions:**

This paper proposes two plug-and-play modifications for self-supervised representation learning, 1) smoothing the target image to make the model focus on the low-frequency part, and 2) replacing the Resize Random Crop with Simple Random Crop to have better foreground coverage. Experiments show the effectiveness of these modifications on multiple baselines.

**Audience:**

Yes

**Broader Impact Concerns:**

No Broader Impact Statement is available.

**Claims And Evidence:**

Yes

**Requested Changes:**

I would recommend adding experiments on low-level downstream tasks to compare the proposed method and existing methods.

**Strengths And Weaknesses:**

Strengths:
1. The proposed method is simple and effective. Experiments are performed on multiple self-supervised learning baselines.
2. The proposed method has a good intuition. The two problem identified is meaningful for this task.
3. The writing is good and the presentation is clear.
4. Code will be released.
5. Ablation studies is good in general.

Weaknesses:
1. Although the proposed method is simple and effective, it might provide enough insights and inspiration from the methodology perspective. They are more like two implementation tricks, which also need hand-tuning. I think it will be more interesting to further extend the proposed method, for example, on how to adaptively select the most informative and helpful patches and frequency components.
2. The smooth target after the low-pass filter has been shown to be useful for high-level downstream tasks. However, I am curious whether it is useful for low-level downstream tasks. Intuitively, low-level downstream tasks will rely more on the high-frequency components than high-level downstream tasks.

---

### Review · Reviewer_47BK · 2023-11-30

**Summary Of Contributions:**

In this paper, the authors examined masked image modeling in detail and found two important bottlenecks of previous methods including the image reconstruction target and the input patch cropping method. It is found that the previous method reconstructed pixels uniformly along all the frequency domains, which might not be necessary. Secondly, the authors also found that the random sized cropping could not cover enough foreground pixels. Based on the two analysis, the authors proposed two simple techniques to improve the performance of MIM methods. First, the high-frequency components are removed from the reconstruction target. So the network do not need to reconstruct these frequencies. Secondly, a simple sized crop method is used to replace the random sized cropping. Expreimental results show the advantage of the proposed techniques.

**Audience:**

Yes

**Broader Impact Concerns:**

I don't think there are any ethical concerns in the field of masked image modeling (MIM). Apart from that, the basic contribution of the paper is the improved training strategy for MIM models. The authors also tried to validate this idea with various MIM methods. Thus, this paper should be interesting to the broader research community.

**Claims And Evidence:**

Yes

**Requested Changes:**

1. Investigating a diversity of models and model size could further improve the paper. Will the benefit of the two proposed strategies diminish as the model size gets much larger?
2. As shown in Figure 5, PixMAE shifts the model's focus to the low-frequency components. But the high-frequency components are not completely removed as the the reconstruction target. Why is that?
3. Will suppressing the low-frequency components further improve the performance of the network?

**Strengths And Weaknesses:**

1. The authors delve into MIM models and find out two bottlenecks in the previous methods. The two points are overlooked in previous works. An investigation into these points shed light on how to improve the training of MIM models.
2 The proposed techniques are simple yet effective. The removing of high-frequency components and using simple sized crop that keep more foreground pixels are straightforward ways to solve the problems mentioned by the authors. It is surprising that the simple modification works quite well.
3. The experimental results support what is claimed in the paper.
4. Sufficient ablation study is done to investigate the influence of the proposed two techniques.

---

### Decision · Action_Editor_3ESy · 2024-01-15

**Recommendation:** Accept with minor revision

**Comment:**

It appears that the authors' answers to the reviewers' feedback were not incorporated in the manuscript. The AE recommends the author to do so, in particular regarding:
- the high-frequency components in Figure 5;
- the effect of suppressing low-frequency components;
- the use of an adaptive filtering threshold;
- the limited impact of PixMIM on methods that use high-semantic features as targets, such as MILAN;
- the use of PixMIM with larger models (already in Table 16, but 2 reviewers missed it, so it might be worth mentioning it more explicitly in the main text).

**Audience:**

All reviewers agree that this work is of interest to the TMLR audience.

**Claims And Evidence:**

The reviewers in general acknowledge the interest of the analysis performed by the authors and the effectiveness of the proposed solutions to address the two weaknesses of MiM identified by this analysis. In their initial reviews, the reviewers nonetheless asked the authors to clarify and evaluate a few aspects of their work. The authors did so mostly convincingly, except for one comment from Reviewer khNT regarding low-level tasks. Altogether, the AE believes that the claims made in this submission are thus indeed supported by clear evidence.